# The countrywide historical gravity dataset of Lithuanian territory

Dominykas Šlikas, Eimuntas Paršeliūnas, Rosita Birvydienė, Romuald Obuchovski Institute of Geodesy, Vilnius Gediminas Technical University, Vilnius, LT-10223, Lithuania *Correspondence to*: Dominykas Šlikas (dominykas.slikas@vilniustech.lt)

- 5 Abstract. The historical gravity dataset of Lithuania consists of two files: data of second and third order gravimetric network as well as data of gravity survey. Raw data were collected by digitising the paper catalogues of gravimetric network stations and sheets of gravimetric map at a scale 1:200000. Gravity data set covers total territory of Lithuania (65 thousand square km). Raw data were collected by digitising the paper catalogues of gravimetric network stations and sheets of gravimetric map at a scale 1:200000. Gravity data set covers total territory of Lithuania (65 thousand square km). Raw data were collected by digitising the paper catalogues of gravimetric network stations and sheets of gravimetric map at a scale 1:200000. The digitiser CALCOMP 9600 was employed along with ARC/INFO software to detect geodetic coordinates
- 10 of gravity points from the map sheets. Initially the gravity data were in Potsdam gravity system, geodetic coordinates of the gravity points in the coordinate reference system Pulkovo 1942 (EPSG:2499), and the heights of gravity points in the Baltic normal height system of 1977 (EPSG code 5705). In the final countrywide set the gravity data are in the International Gravity Standardization Net of 1971 gravity system, geodetic coordinates in European Terrestrial Reference System of 1989 coordinate reference system (EPSG:4258), and geodetic heights in the European Vertical Reference System of 2007
- 15 (EPSG:5215). Total number of gravimetric network stations is 123, and total number of gravity survey points is 10660. The data were recorded into files applying DBF (Data Base Format) format. Historical gravity data set could be used for quasi-geoid modelling and for development of Earth geopotential models. Researchers will benefit in the process of evaluation and accuracy estimation of developed products using high precision data of gravity network stations.

# **1** Introduction

- 20 The historical gravity survey of the Earth's gravity field in Lithuania territory was carried out in 1951–1962 (Paršeliūnas and Petroškevičius, 2007; Paršeliūnas et al., 2010; Petroškevičius, 2004; Petroškevičius et al., 2014). The reference gravity stations in Vilnius, Panevėžys, Rīga, Daugavpils, Lida and Karaliaučius were used. In total 10660 gravimetric points were observed. On the basis of this gravity survey the gravimetric map at a scale 1:200 000 was generated. The gravity data was in Potsdam gravity system, geodetic coordinates of the gravity points in the coordinate reference system Pulkovo 1942 (EPSG:2499),
- and the heights of gravity points in the Baltic normal height system of 1977 (EPSG code 5705). Investigations showed that the average accuracy of the gravity due acceleration, detected from gravimetric map, is about 0.7 mGal. However, in some areas accuracy is much worse and goes down till 3 mGal (Birvydienė, 2010). The gravimetric network of the second (18 stations) and third order (105 stations) was developed as well (Paršeliūnas and Petroškevičius, 2007; Petroškevičius, 2004). The overview of historical gravity data set is presented in Figure 1.