# Peer review of "The countrywide historical gravity dataset of Lithuanian territory"

_Earth System Science Data, 2025_

## Referee Comment (RC1)

**Review**
**Title:** The countrywide historical gravity dataset of Lithuanian territory
**Authors:** Dominykas Šlikas and colleagues

**General Assessment**
This manuscript presents a digitized and homogenized historical gravity dataset for the territory of Lithuania, incorporating data from mid-20th century gravity surveys and network stations. The dataset has been transformed into modern geodetic and gravimetric reference systems (ETRS89, EVRS07, IGSN71) and is made publicly available via Zenodo. This is a valuable contribution to the field, particularly for applications in regional geoid modeling and Earth gravity field studies.

The manuscript should be published as soon as possible, but only after substantial revision in both its presentation and clarity to meet the standards of *Earth System Science Data.*

**Major comments**

**Scientific contribution and novelty**
The authors provide a robust and well-documented compilation of historical gravity data. The strength of the work lies in data preservation and accessibility, not in methodological innovation. The manuscript does not introduce any new methodology; data processing follows standard transformation and reduction formulas. This should be clearly acknowledged in the manuscript.

**Use of IGSN71**
The "International Gravity Standardization Net of 1971" (IGSN71) is no longer considered the most accurate or modern reference for gravity data. While still widely used for legacy compatibility (e.g., in some global gravity field models), newer systems based on absolute gravimetry and regional densifications are becoming the standard, especially in Europe.

The EUREF sub-commission, which also maintains ETRS89 and EVRS, recommends the use of absolute gravity values where available, referencing new gravity campaigns to co-located absolute stations, and avoiding continued reliance on IGSN71 unless necessary for legacy data comparison.

**My suggestion:** The authors should justify their use of IGSN71 and add two statements such as:
"While IGSN71 is used here to harmonize legacy gravity data, future work should integrate absolute gravity stations in the European Absolute Gravity Reference Network."
"If possible, transformation offsets to newer national absolute reference values should be computed and documented."

**Clarity and structure of manuscript**
The manuscript is overly long and detailed, with extended theoretical derivations (e.g., coordinate transformations, gravity reductions) that could be summarized or moved to supplementary material. Repetition occurs throughout (e.g., multiple restatements of the Potsdam to IGSN71 transformation), and the language would benefit significantly from professional editing for grammar, conciseness, and flow. At least for me, the tone is at times more instructional than scientific.

**Figures and maps – major weakness**
The figures are currently a critical weak point:
- Several maps lack coordinates (e.g., Figs. 1, 2.1, 2.4, 2.8, 2.9, 2.10), scale bars, or consistent legends.
- Different map extents and projections are used without explanation (e.g., Figs. 2.2, 2.3, 2.4–2.7), making comparison difficult.
- Important metadata (e.g., color meanings, units) are missing or unclear.
- The layout and visual clarity are insufficient for a data-focused journal.

**My recommendation:** All figures, particularly maps, should be redesigned for standardization. Use the same geographic extent, add graticules or coordinate grids, include legends and scale bars, and clearly distinguish between data types (e.g., second/third order stations, gravity points). Figures 2.1, 2.4, and 2.10 could be consolidated or at least harmonized.

**Relevance and utility of the dataset**

The dataset is well motivated, and its availability via a public repository is appropriate. The data will be useful for researchers in regional geoid computation, historical geophysics, and geodetic accuracy assessment.

A brief demonstration of a possible application, such as geoid height comparison or regional gravity model validation, would enhance the manuscript's impact.

**Data format**

The authors state (Section 3) that the dataset is published in DBF (dBASE) format, which is technically valid but increasingly outdated. While DBF is a well-documented and stable format, it lacks support for Unicode, long field names, and complex data types. It is also not readily used in modern data processing environments without conversion.

**My suggestion:** While DBF is functional, I encourage the authors to consider releasing the dataset in additional formats, such as CSV, GeoPackage, or NetCDF, which are more accessible, transparent, and compatible with current data processing tools. This would significantly enhance the reusability of the dataset.

**Minor comments and suggestions**

- Replace "due gravity" with "due to gravity" or simply "gravity" throughout the manuscript.
- The abstract could be made more concise.
- Section 2.2 would benefit from a table or flowchart summarizing the transformation steps.
- The DBF table (Section 3) is useful, but a sample dataset or a visual preview (e.g., GIS screenshot) would help readers better understand the structure.
- Include a graphical summary of transformation errors (e.g., histogram of residuals).
- Use SI units consistently and replace "mGal" with $m/s^2$.
- References are mostly appropriate, but citation style and hyperlinks should be carefully checked and standardized.

**Overall recommendation: Accept with major revisions.**

---

## Referee Comment (RC2)

**The countrywide historical gravity dataset of Lithuanian territory**

Dominykas Šlikas, Eimuntas Paršeliūnas, Rosita Birvydienė, Romuald Obuchovski

Institute of Geodesy, Vilnius Gediminas Technical University, Vilnius, LT-10223, Lithuania

*Correspondence to*: Dominykas Šlikas (dominykas.slikas@vilniustech.lt)

5   **Abstract.** The historical gravity dataset of Lithuania consists of two files: data of second and third order gravimetric network as well as data . Raw data were collected by digitising the paper catalogues of gravimetric network stations and sheets of gravimetric map at a scale 1:200000. Gravity data set covers total territory of Lithuania (65 thousand square km). Raw data were collected by digitising the paper catalogues of gravimetric network stations and sheets of gravimetric map at a scale 1:200000.

10   . Initially the gravity data were in Potsdam gravity system, geodetic coordinates of the gravity points – in the coordinate reference system Pulkovo 1942 (EPSG:2499), and the heights of gravity points – in the Baltic normal height system of 1977 (EPSG code 5705). In the final countrywide set the gravity data are in the International Gravity Standardization Net of 1971 gravity system, geodetic coordinates – in European Terrestrial Reference System of 1989 coordinate reference system (EPSG:4258), and geodetic heights – in the European Vertical Reference System of 2007

15   (EPSG:5215). Total number of gravimetric network stations is 123, and total number of gravity survey points is 10660. The data were recorded into files applying DBF (Data Base Format) format. Historical gravity data set could be used for quasi-geoid modelling and for development of Earth geopotential models. Researchers will benefit in the process of evaluation and accuracy estimation of developed products using high precision data of gravity network stations.

**1 Introduction**

20   The historical gravity survey of the Earth's gravity field in Lithuania  was carried out in 1951–1962 (Paršeliūnas and Petroškevičius, 2007; Paršeliūnas et al., 2010; Petroškevičius, 2004; Petroškevičius et al., 2014). The reference gravity stations in Vilnius, Panevėžys, Rīga, Daugavpils, Lida and Karaliaučius . In total 10660 gravimetric points were observed. On the basis of this gravity survey the gravimetric map at a scale 1:200 000 was generated.  Potsdam gravity system, geodetic coordinates of the gravity points – in the coordinate reference system Pulkovo 1942 (EPSG:2499),

25   and the heights of gravity points – in the Baltic normal height system of 1977 (EPSG code 5705). Investigations showed that the average accuracy of the gravity  from gravimetric map, is about 0.7 mGal. However, in some areas accuracy is much worse  3 mGal (Birvydienė, 2010). The gravimetric network of the second (18 stations) and third order (105 stations) was developed as well (Paršeliūnas and Petroškevičius, 2007; Petroškevičius, 2004). The overview of historical gravity data set is presented in Figure 1.

[Figure]

[Figure]

30

[Figure]

**Figure 1: Distribution of the historical gravimetric points in the territory of investigation and compilation. Colours indicate the types of map sheets used in the compilation** (see Figure 5).

Similar gravity data sets  compiled at countrywide (APAT, 2005; Basic and Bjelotomic, 2014; Csapó et al., 2000; Martelet et al., 2009; Meurers and Ruess, 2009; Stopar, 2016), regional (Zahorec, 2020; Bielik et al., 2006; Denker and Torge, 1998;

35 Denker and Roland, 2005; Ågren et al., 2016) and global scales (Förste et al., 2014; Pavlis et al., 2012).

**2 Experimental design, materials and methods**

**2.1 Historical gravity data sources**

The historical gravity data set is based on two gravity data sources:

- Second and third order gravity network stations (18 and 105 gravimetric stations);

40 - Gravimetric  at a scale 1:200 000 (10 660 gravimetric points).

The data of the second and third order gravity network stations were collected by digitising the catalogues of the gravimetric stations. Distribution of the second and third order gravimetric stations in the territory of Lithuania is presented in the Figure 2.

[Figure]

**Figure 2: Distribution of the second and third order gravimetric stations (green dots – second order stations; magenta dots – third order stations).**

The gravimetric measurements of the acceleration due gravity were carried out by relative gravimeters ГАК-3М, ГАК-4М, ГАК-ПТ, ГАК-7Т. The standard deviation of the accelerations due to gravity does not exceed 0.35 mGal (Birvydienė, 2010). The coordinate reference system of Pulkovo 1942 (EPSG code 2499), the Krassowsky ellipsoid of 1940 (EPSG code 7024) and Baltic normal height system of 1977 (EPSG code 5705) were used for positioning of gravimetric points. The accelerations due gravity were expressed in Potsdam system and applying Helmert normal gravity field.

The gravimetric map was based on data from the gravimetric survey carried out in period from 1951 till 1962 (Birvydienė, 2010). Gravimetric measurements of the accelerations due to gravity were performed by relative gravimeters СН-3, ГКА, ГАК-3М, ГАК-4М, ГКМ. Normal heights of the gravimetric points were measured by geometric - barometric levelling or detected from topographic map at a scale of 1:25000. The third class gravimetric network was used as base for the map. The coordinate reference system of Pulkovo 1942 (EPSG code 2499), the Krassowsky ellipsoid of 1940 (EPSG code 7024) and Baltic height system of 1977 (EPSG code 5705) were used for positioning of gravimetric points. On the basis of this gravity survey, the gravimetric map at a scale 1:200 000 was constructed applying Gauss-Krüger projection (EPSG code 2499). The fragments of the gravimetric map are presented in the Figure 3 and 4.

[Figure]

60    **Figure 3:**  of the gravimetric map at a scale 1:200 000. (The firm, 1965).

[Figure]

**Figure 4: Fragment of the gravimetric map at a scale 1:200 000 containing the Bouguer anomalies and normal heights of gravimetric points. (The firm, 1965).**

The Bouguer anomalies due to gravity were written at each point in the map. The total number of gravimetric points is 10 663.

The Helmert's formula for the estimation of normal gravity field and the density of the Earth's crust $\delta = 2.3\ g \cdot cm^{-3}$ were used for calculations of the anomalies due gravity. The gravity field is presented by isolines at every 2 mGal. The Bouguer anomalies due to gravity were calculated following Eq. (1) (Torge, 1989):

$$\Delta g_P^B = g_P - \gamma_H^0 + 0.3086 H_{77} - 0.0419\,\delta \cdot H_{77}, \tag{1}$$

where $g_P$– Potsdam system gravity acceleration measured at the point on the Earth surface; $\gamma_H^0$– the acceleration of Helmert normal gravity field on equipotential ellipsoid surface, $H_{77}$ - normal height at point on the Earth's surface in the Baltic Height System of 1977. The accelerations of Helmert's normal gravity field on equipotential ellipsoid surface were obtained following Eq. (2) (Petroškevičius, 2004):

[Figure]

$$\gamma_H^0 = 978030\,(1 + 0.005302\,sin^2\,B_{42} - 0.000007\,sin^2\,2\,B_{42}), \tag{2}$$

where $B_{42}$ – the latitude of gravimetric point in Pulkovo 1942 coordinate reference system.

75  Unfortunately the number of the map sheets containing normal heights of the gravimetric points was very limited (Fig. 5).

**Figure 5:** **Types of map sheets to detect the normal heights of the gravimetric points.**  indicate the types of map sheets used in the compilation: green – map sheets containing the normal heights; blue – normal heights derived from Bouguer anomalies and free air anomalies; red – normal heights interpolated from topographic map.

80  To  the normal heights of gravimetric points :

   • Paper map sheets at a scale 1:200 000  free air anomalies  (Fig. 6) leading by maps sheets, containing the normal heights of gravimetric points (Fig. 7).

- Paper map sheets of the topographic map at a scale 1:10 000 (Fig. 8).

[Figure]

85 **Figure 6: Fragment of the map sheet containing the free air anomalies due gravity. (The firm, 1965).**

[Figure]

[Figure]

**Figure 7:  map sheet containing the normal heights of gravimetric points. (The firm, 1965).**

[Figure]

[Figure]

**Figure 8:**  topographic map sheet at a scale 1:10 000. (Topographic, 1958)

**2.2 Modern compilation of the historical gravity data set**

 compilation of the historical gravity data set includes three items:

- Introduction the European Terrestrial Coordinate System of 1989 (ETRS89).
- Introduction the European Vertical Reference System of 2007 (EVRS07).
- Introduction the International Gravity Standardization Net of 1971 gravity system (IGSN71).

**2.2.1 Introduction the coordinate reference system ETRS89**

One of the key problems in the unification of gravimetric databases is the homogenization of position, height and gravimetric coordinate systems used in each database. Through its historical development, each country has used and sometimes still uses

local systems and their realisation (frame), which are often based on the established principles of reference systems using older ellipsoids or older geodetic reference networks and projections. These systems and their realisations thus contain several

100    differences, which are responsible for large inhomogeneities, shifts, errors in position, height, and gravity. These errors are most evident in the mutual comparison of data from individual countries.

To avoid these problems in the position of gravimetric points, all position data were transformed from local systems to the global system, i.e., the European Terrestrial Reference System 1989, which is accurate, homogeneous, and recommended for all European countries (Altamimi and Collilieux, 2024). A similar situation is in the height systems where countries use

105    different types of physical heights, they are linked to different tide gauges and each country has a different practical implementation of the relevant height system (EVRS, 2020). The solution is again transformation to a uniform platform in the form of ellipsoidal heights in the ETRS89 system based on the ellipsoid GRS80 (Moritz, 1984; Moritz, 2000). The situation is similar in gravimetric reference systems, where especially the gravimetric databases that have been created for decades often use old gravimetric systems connected to the Potsdam system. An important step was therefore to convert these data into

110    gravimetric systems, which are connected to absolute gravimetric points and measurements, such as IGSN71 (Morelli et al., 1972) or modern national systems connected with the recent absolute measurements, which are verified by international comparisons of absolute gravimeters (Francis et al., 2015).

The Coordinate Reference System of gravimetric points both from the catalogues and from gravimetric map was Pulkovo 1942 system (EPSG code: 2499). To express the geodetic coordinates of the points in modern European Terrestrial Reference System

115    of 1989 (ETRS89, EPSG code: 4258) the three steps algorithm was employed. In the first step the geocentric coordinates of the gravimetric points were calculated from the geodetic ellipsoidal coordinates expressed in Pulkovo 1942 system applying formulas from (Schödlbauer, 1981; Schödlbauer, 1982; Annoni et al., 2001).

Algorithm. Conversation the ellipsoidal coordinates to the rectangular geocentric coordinates.

Input. The ellipsoidal coordinates: geodetic latitude $B_{42}$, geodetic longitude $L_{42}$, ellipsoidal height $H_e^K$ and parameters of

120    ellipsoid a and f (in our case – Krassowsky 1940 ellipsoid: a=6378245 m, f=298.3).

Output. The rectangular geocentric coordinates $X'_{42}, Y'_{42}, Z'_{42}$.

Calculation formulaes (CRS-Geo, 2025):

$$X'_{42} = (N_K + H_e^K) \cos B_{42} \cos L_{42}, \tag{3}$$

$$Y'_{42} = (N_K + H_e^K) \cos B_{42} \sin L_{42}, \tag{4}$$

125    $$Z'_{42} = (N_K + H_e^K - N_K e^2) \sin B_{42}, \tag{5}$$

where

$$N_K = \frac{a}{\sqrt{1 - e^2 \sin^2 B_{42}}}, \tag{6}$$

$$e^2 = 2 \cdot f - f^2. \tag{7}$$

[Figure]

[Figure]

**Example.** The Pulkovo 1942 (Krassowsky 1940 ellipsoid) ellipsoidal coordinates of gravimetric point are: B42 = 55° 15'

130    22.9", L42 = 23° 52' 26.1", $H_e^K$ = 22.4 m. Calculated rectangular geocentric coordinates are:

$X'_{42}$ = 3331529.1 m, $Y'_{42}$ = 1474516.4 m, $Z'_{42}$ = 5217810.9 m.

In the second step the geocentric coordinates of the gravimetric points expressed in Pulkovo 1942 system were transformed to

ETRS89 geocentric coordinates applying standard Helmert's seven parameters transformation formula (Reit, 2009)

$$\begin{pmatrix} X \\ Y \\ Z \end{pmatrix} = \begin{pmatrix} \delta x \\ \delta y \\ \delta z \end{pmatrix} + R \begin{pmatrix} X' \\ Y' \\ Z' \end{pmatrix} (1 + \delta m), \tag{8}$$

135    where $X_{89}, Y_{89}, Z_{89}$ – rectangular geocentric coordinates of the points in ETRS89 system, $X'_{42}, Y'_{42}, Z'_{42}$ – rectangular

geocentric coordinates of the same points in Pulkovo 1942 system, R – matrix of rotations:

$$R = \begin{pmatrix} 1 & +\omega_z & -\omega_y \\ -\omega_z & 1 & +\omega_x \\ +\omega_y & -\omega_x & 1 \end{pmatrix}. \tag{9}$$

Values of the transformation parameters between Pulkovo 1942 and ETRS89 systems were derived by analysing the geocentric

coordinates of 45 geodetic points, evenly distributed  (Fig. 9).

[Figure]

140

**Figure 9: Distribution of geodetic points used for the transformation parameters calculation**

Values of the transformation parameters applied to geodetic coordinates of the gravimetric points  gravity data set:

$\delta x$ = -40.5953 m;

$\delta y$ = -18.5498 m;

145    $\delta z$ = -69.3396 m;

[Figure]

$\omega_x$ = -0.0000121590 rad;

$\omega_y$ = -0.0000088812 rad;

$\omega_z$ = +0.0000126606 rad;

$\delta m$ = -0.0000042991.

150    **Example.** Input: geocentric coordinates in CRS Pulkovo 1942:

$X'_{42}$= 3331529.1 m, $Y'_{42}$ = 1474516.4 m, $Z'_{42}$ = 5217810.9 m.

Output: geocentric coordinates in CRS ETRS89:

$X_{89}$ = 3331567.8 m, $Y_{89}$ = 1474398.6 m, $Z_{89}$ = 5217752.3 m

In the third step the ETRS89 ellipsoidal coordinates were calculated from the ETRS89 geocentric coordinates applying

155    formulas from (Schödlbauer, 1981; Schödlbauer, 1982; Annoni et al., 2001):

**Algorithm.**  the rectangular geocentric coordinates to the ellipsoidal coordinates.

Input. The rectangular geocentric coordinates $X_{89}, Y_{89}, Z_{89}$ and parameters of ellipsoid a and f (in our case parameters of GRS80 ellipsoid, a= 6378137 m, f =298.257222101 (Moritz, 1984; Moritz, 2000).

Output. The ellipsoidal coordinates: geodetic latitude $B_{89}$, geodetic longitude $L_{89}$ and ellipsoidal height $H_e^{80}$.

160    Calculation formulaes:

$$L_{89} = arctg(Y_{89}/X_{89}); \tag{10}$$

Initial geodetic latitude:

$$B_0 = arctg(Z_{89}/ ((1 - e_{80}^2) (X_{89}^2 + Y_{89}^2))^{1/2}); \tag{11}$$

where

165    $$e_{80}^2 = 2 \cdot f - f^2. \tag{12}$$

Iterations till $B_i - B_{i-1} > \Delta$, where $\Delta$ is appropriately chosen value defining the necessary precision of latitude $B$ (in our case $\Delta = 0.1''$):

$$N_i = \frac{a}{\sqrt{1 - e_{80}^2 \sin^2 B_{i-1}}}; \tag{13}$$

$$H_i^e = \frac{Z}{\sin B_{i-1}} - (1 - e_{80}^2)N_i, \tag{14}$$

170    $$B_i = arctg\left(\frac{Z_{89}}{(X_{89}^2 + Y_{89}^2)^{1/2}} \cdot \frac{1}{1 - (e_{80}^2 N_i)/(N_i + H_i^e)}\right). \tag{15}$$

**Example.** The rectangular geocentric coordinates of gravimetric point are:

$X_{89}$ = 3331567.8 m, $Y_{89}$ = 1474398.6 m, $Z_{89}$= 5217752.3 m.

Calculated ETRS89 (GRS80 ellipsoid) ellipsoidal coordinates are:

[Figure]

$B_{89} = 55°15'22.2"$, $L_{89} = 23°52'19.1"$ , $H_e^{80} = 77.1$ m.

**2.2.2 Introduction the European Vertical Reference System EVRS07**

EVRS07 (EPSG: 5215) is realized by geopotential numbers and normal heights of the United European Levelling Network (UELN) (Sacher et al., 2008; Ihde et al., 2008; Dragomir et al., 2011). The newest realisation of EVRS is EVRF2019 (EPSG: 1274). EVRF2019 is a zero-tide surface (Sacher and Liebsch, 2019). It should be noted, that difference between two realisation in Lithuania territory is about 1 cm only and for calculations of histrical gravity data set is negligible. Federal Agency for Cartography and Geodesy (BKG – Bundesamt fűr Kartographie und Geodesie), Germany, and the Reference Frame Sub-Commission for Europe (EUREF) have developed a formulae for transformation between the local height (vertical) systems and European Vertical Reference System of 2007 (Celms et al., 2014; Kadaj, 2018; Dragomir et al., 2011). In the historical gravity data set the normal heights of gravity points were expressed in Baltic Heights System of 1977 referred to tide gauge Kronstadt (BHS77). Formulae to transform BHS77 normal height to EVRS07 normal height is as follow Eq. (16):

$$H_{07} = H_{77} + \alpha_1 + \alpha_2 M_0 (B_{89} - B_0) + \alpha_3 N_0 (L_{89} - L_0) cos B_{89}, \tag{16}$$

where

$H_{07}$ – normal height referred to EVRS07 vertical system;

$H_{77}$ – normal height referred to BAS77 height system;

$P_0(B_0, L_0)$ – origin for the height transformation in Lithuania with geodetic latitude and longitude in ETRS89 CRS;

$M_0$ – radius of curvature in the meridian at the point $P_0$;

$N_0$ – radius of curvature of the prime vertical at the point $P_0$;

$\alpha_1$ – coefficient of the displacement in vertical direction;

$\alpha_2$ – coefficient of the slope along meridian;

$\alpha_3$ – coefficient of the slope along prime vertical;

$B_{89}$ – latitude of the point to transform;

$L_{89}$ – longitude of the point to transform;

The formulas for $M_0$ and $N_0$ can be found in number of sources, for example (Tobler, 1964; Lenart, 2013; Lenart, 2017):

$$N_0 = \frac{a}{\sqrt{(1 - e_{80}^2 sin^2 B_0)}};$$

$$M_0 = \frac{a(1 - e_{80}^2)}{(1 - e_{80}^2 sin^2 B_0)^{3/2}} = \frac{N_0(1 - e_{80}^2)}{1 - e_{80}^2 sin^2 B_0}$$

where $a$ – semimajor axis and $e_{80}^2$ – squared first eccentricity of GRS80 ellipsoid (Eq. 12).

Distribution of the common points used for calculation of the transformation parameters is shown in Fig. 10.

[Figure]

[Figure]

**Figure 10. Distribution of the common points used for calculation of the transformation parameters (233 points)**

In total there are 233 height benchmarks in  Lithuania for which normal heights could be found in both height

205  systems. Parameters of transformation between BHS77 and EVRS07 were received as follow:

$B_0 = 55°18'$ ; $L_0 = 24°01'$; $\alpha_1 = 0.1425$ m; $\alpha_2 = 0.06076"$; $\alpha_3 = 0.03375"$.

The minimum value of residuals equal to -0.025 m and maximum value equal to +0.024 m were obtained. The standard

deviation of transformation equal to 0.013 m was derived.

**Example:**

210  $B_{89} = 55°42'$ ; $L_{89} = 24°21'$; $H_{77} = 55.90$ m ; $H_{07} = 56.06$ m.

**2.2.3 Introduction the Gravity system IGSN71**

The values of accelerations due to gravity $g_p$, digitised from catalogues and referred to the Potsdam gravity system were

recalculated to the IGSN71 system ($g_{71}$) by simple deduction of 14.0 mGal as a difference between two gravity systems

(Wollard, 1979; Torge, 1989; Petroskevicius, 2004):

215  $g_{71} = g_p - 14.0,$                                                                   (17)

Further, the free air anomalies due to gravity were calculated following Eq. (18):

$\Delta g_{71}^F = g_{71} - \gamma_{80}^0 - \Delta\gamma,$                                                (18)

where $\gamma_{80}^0$ – the accelerations of GRS80 normal gravity field on equipotential ellipsoid surface:

[Figure]

$$\gamma_{80}^0 = \gamma_{80e}^0 \frac{1 + k_{80} \sin^2 B_{89}}{\sqrt{1 - e_{80}^2 \sin^2 B_{89}}}, \tag{19}$$

220    where $\gamma_{80e}^0$, $e_{80}$, $k_{80}$ – parameters of GRS80 normal gravity field (Moritz, 2000); $\gamma_{80e}^0$ = 978032.67715 mGal;

$e_{80}^2$ = 0.00669438002290; $k$ = 0.001931851353; $B_{89}$ – latitude of gravimetric point in ETRS89 system.

Corrections due normal heights $H_{07}$ were calculated following Eq. (20):

$$\Delta\gamma = -0.30855(1 + 0.00071\,cos(2B_{89}))H_{07}. \tag{20}$$

Standard deviations of free air gravity anomalies were calculated following Eq. (21):

225    $$m_l = \sqrt{m_g^2 + m_{\gamma^0}^2 + m_{\Delta\gamma}^2}, \tag{21}$$

where $m_g$ - standard deviation of measured gravity acceleration (from the catalogue), $m_{\gamma^0}$ - standard deviation of normal gravity acceleration ($m_{\gamma^0} = 1.4m_B'$, $m_B'$ - standard deviation of geodetic latitude in minutes), $m_{\Delta\gamma}$ - standard deviation of free air reduction ($m_{\Delta\gamma} = 0.30855\,m_H$, $m_H$ – standard deviation of normal height in meters).

Bouguer anomalies due to gravity were calculated following Eq. (22):

230    $$(\Delta g_{71}^B)_{2.67} = \Delta g_{71}^F - \Delta g_{2.67}, \tag{22}$$

where correction for  was calculated following Eq. (23):

$$\Delta g_{2.67} = -2\pi \cdot G \cdot \delta \cdot H_{07}, \tag{23}$$

where Newtonian gravitational constant $G = 6.6743 \cdot 10^{-8} cm^3 g^{-1} s^{-2}$, density of the Earth's crust $\delta = 2.67\ g \cdot cm^{-3}$.

**Example:** $g_p = 981552.8$ mGal ; $H_{07} = 125.0$ m ; $B_{89} = 55°59.3'$ ; $g_{71} = 981538.8$ mGal ; $\Delta g_{71}^F = -13.67$ mGal ;

235    $(\Delta g_{71}^B)_{2.67} = -27.67$ mGal; $m_B = 0.1'$; $m_H = 0.5$ m; $m_l = 0.7$ mGal.

 the values of accelerations due to gravity in IGSN71 system were calculated following Eq. (24):

$$g_{71} = (\Delta g_P^B)_{2.3} + \gamma_H^0 - \Delta\gamma - \Delta g_{2.3} - 14.0, \tag{24}$$

where $(\Delta g_P^B)_{2.3}$ – Bouguer anomalies due gravity digitised from gravimetric map sheets and referred to the Potsdam gravity system, $\gamma_H^0$ – normal gravity value, $\Delta\gamma$ – correction due normal height (Eq. 20), $\Delta g_{2.3}$ – correction for

240    :

$$\gamma_H^0 = 978030 \cdot (1 + 0.005302 sin^2 B_{89} - 0.0000070 sin^2(2B_{89})), \tag{25}$$

$$\Delta g_{2.3} = -0.0419277 \cdot \delta \cdot H_{07}, \tag{26}$$

where density of the Earth's crust $\delta = 2.3\ g \cdot cm^{-3}$.

[Figure]

In case when the normal heights of the gravimetric points are unknown, the normal heights in BHS77 system were calculated

245    from values of free air anomalies and Bouguer anomalies following Eq. (27):

$$H_{77} = [\Delta g_P^F - (\Delta g_P^B)_{2.3}]/0.09637 \tag{27}$$

where $\Delta g_P^F$ – free air anomaly  (from gravimetric map of free air gravity anomalies ), $(\Delta g_P^B)_{2.3}$ –
Bouguer anomaly  (from gravimetric map of Bouguer anomalies due gravity).

**Example**: $\Delta g_P^F = 4.1$ mGal; $(\Delta g_P^B)_{2.3} = 3.1$ mGal; $H_{77} = 10.4$ m.

250    Further, the free air and Bouguer anomalies  were calculated following Eq. (18 and 22).

**Example**: $B_{89} = 55°59.1'$; $H_{07} = 10.4$ m; $(\Delta g_P^B)_{2.3} = 3.1$ mGal; $\Delta g_{71}^F = -13.8$ mGal; $(\Delta g_{71}^B)_{2.67} = -14.99$ mGal; $m_{Bg} = 0.7$ mGal; $m_H = 0.5$ m; $m_l = 0.7$ mGal.

Figure 11 shows the spatial distribution of gravimetric points of all countrywide historical gravity data set.

[Figure]

255    **Figure 11.** Spatial distribution of gravimetric points of all historical gravity data set (green dots – second order stations; magenta dots – third order stations; black dots – gravity survey points).

**3 Data availability**

The historical gravity data set is available in the ZENODO repository at https://doi.org/10.5281/zenodo.15090241 (Šlikas et al., 2025). The data were recorded into files applying DBF (Data Base Format) format (Digital, 2012). The database

260    structure is presented in Table.

The database structure of the historical gravity data set

[Figure]

| No. | Field name | Field type | Data format | Description | Example |
|-----|-----------|-----------|-------------|-------------|---------|
| 1 | PNUMB | N | 8 | Point number | 100002 |
| 2 | B_89 | N | 8,5 | ETRS89 latitude, degrees | 55.330333 |
| 3 | L_89 | N | 8,5 | ETRS89 longitude, degrees | 21.342667 |
| 4 | HN_07 | N | 6,2 | EVRS07 normal height, m | 1.04 |
| 5 | GA_71 | N | 9,2 | IGSN71 Gravity acceleration, mGal | 981510.96 |
| 6 | FAA | N | 6,2 | Free air anomaly due to gravity, mGal | -24.24 |
| 7 | ABUG_27 | N | 6,2 | Bouguer anomaly due to gravity, mGal | -24.36 |
| 8 | GA_RMSE | N | 4,2 | Root mean square error of gravity acceleration, mGal | 0.70 |

**4 Conclusions**

1. In this study, the countrywide historical gravity data set was compiled. It consists of two database files: data of gravity network points (123 stations) and data of gravity survey (10660 points). The estimated accuracy of gravity network stations is about 0.2 mGal (second order stations) and 0.35 mGal (third order stations), and accuracy of the gravity survey points is about 0.7 mGal.

2. The transformation parameters and algorithms to introduce the European Terrestrial Coordinate System of 1989 (ETRS89), European Vertical Reference System of 2007 (EVRS07) and International Gravity Standardization Net of 1971 gravity system (IGSN71) were defined.

3. These data are useful in understanding the gravity field of territory of the entire country and could be used for quasi-geoid modelling and creating the geopotential models. Researchers will benefit during evaluation and accuracy estimation of developed products using high precision data of gravity network points.

**Author contributions.** All the authors contributed to recovering historical gravity data set and editing the manuscript. EP and RO designed the study, writing the manuscript. RB and DS did analysis and interpretation of the data. DS did compiled DBF files.

**Competing interests.** The contact author has declared that none of the authors has any competing interests.

**Acknowledgements.** We are very grateful to National Land Service under Ministry of Environment for providing the historical gravity maps.

**References**

Ågren, J. et al.: The NKG2015 gravimetric geoid model for the Nordic-Baltic region, https://doi.org/10.13140/RG.2.2.20765.20969, 2016.

[Figure]

Altamimi, Z. and Collilieux, X.: *EUREF Technical Note 1: Relationship and Transformation between the International and the European Terrestrial Reference Systems*, Institut National de l'Information Géographique et Forestière (IGN), France, 15 pp., 2024.

285  Annoni, A., Luzet, C., Gubler, E., and Ihde, J. (Eds.): *Map projections for Europe*, Institute for Environment and Sustainability, 132 pp., 2001.

APAT: Gravity Map of Italy and Surrounding Seas, 1:1 250 000, Agenzia per la protezione dell'ambiente e per Iservizi tecnici, 2005.

Basic, T. and Bjelotomic, O.: HRG2009: New High Resolution Geoid Model for Croatia, in: *Gravity, Geoid and Height*

290  *Systems*, IAG Symposia, 141, 187–191, Springer, https://doi.org/10.1007/978-3-319-10837-7_24, 2014.

Bielik, M., Kloska, K., Meurers, B., Švancara, J., Wybraniec, S., and CELEBRATION 2000 Potential Field Working Group: Gravity anomaly map of the CELEBRATION 2000 region, *Geologica Carpathica*, 57(3), 145–156, 2006.

Birvydienė, R., Krikštaponis, B., Obuchovski, R., Paršeliūnas, E., Petroškevičius, P., and Šlikas, D.: Evaluation of the gravimetric map of Lithuanian territory, *Geodesy and Cartography*, 36(1), 20–24, https://doi.org/10.3846/gc.2010.03, 2010.

295  Celms, A., Bimane, I., and Reke, I.: European Vertical Reference System in Baltic Countries, *Baltic Surveying*, 1, 49–55, 2014.

CRS-Geo: Description of national Coordinate Reference Systems of European Countries, https://www.crs-geo.eu/crs-national.htm, last access: 11 February 2025.

Csapó, G. and Völgyesi, L.: Hungary's new gravity base network (MGH-2000) and its connection to the European Unified

300  Gravity Net, in: *Vistas for Geodesy in the New Millennium*, IAG Symposia, 125, 72–77, Springer, 2000.

Denker, H. and Torge, W.: The European gravimetric quasigeoid EGG97 – An IAG supported continental enterprise, *IAG Symposia*, 119, 249–254, Springer, 1998.

Denker, H. and Roland, M.: Compilation and evaluation of a consistent marine gravity data set surrounding Europe, *IAG Symposia*, 128, 248–253, Springer, 2005.

305  Digital Preservation: dBASE Table File Format (DBF), https://www.loc.gov/preservation/digital/formats/fdd/fdd000325.shtml, last access: 20 March 2025.

Dragomir, P. I., Tiberiu, R., Neculai, A., and Dumitru, P.: EVRF2007 as Realization of the European Vertical Reference System (EVRS) in Romania, *RevCAD Journal of Geodesy and Cadastre*, 1, 51–63, 2011.

Förste, C., Bruinsma, S. L., Abrikosov, O., Lemoine, J. M., Marty, J. C., Flechtner, F., Balmino, G., Barthelmes, F., and

310  Biancale, R.: EIGEN-6C4 – The latest combined global gravity field model including GOCE data up to degree and order 2190 of GFZ Potsdam and GRGS Toulouse, *GFZ Data Services*, https://doi.org/10.5880/icgem.2015.1, 2014.

Ihde, J., Mäkinen, J., and Sacher, M.: Conventions for the Definition and Realization of a European Vertical Reference System (EVRS) – EVRS Conventions 2007, https://www.researchgate.net/publication/265823560, 2008.

Kadaj, R.: Transformations between the height reference frames: Kronsztadt'60, PL-KRON86-NH, PLEVRF2007-NH,

315  *Journal of Civil Engineering, Environment and Architecture*, 65(3), 5–24, https://doi.org/10.7862/rb.2018.38, 2018.

[Figure]

Lenart, A. S.: Solutions of Inverse Geodetic Problem in Navigational Applications, *TransNav*, 7(2), 253–257, https://doi.org/10.12716/1001.07.02.13, 2013.

Lenart, A. S.: Sphere-to-Spheroid Comparison – Numerical Analysis, *Polish Maritime Research*, 24, 4–9, https://doi.org/10.1515/pomr-2017-0129, 2017.

320  Martelet, G., Pajot, G., and Debeglia, N.: Nouvelle carte gravimétrique de la France, RCGF09 – Réseau et Carte Gravimétrique de la France, *Rapport BRGM/RP-57908-FR*, 77 pp., 2009.

Meurers, B. and Ruess, D.: A new Bouguer gravity map of Austria, *Austrian Journal of Earth Sciences*, 102, 62–70, 2009.

Morelli, C., Gantar, C., Honkasalo, T., McConnell, R. K., Tanner, I. G., Szabo, B., Uotila, U., and Whalen, C. T.: The International Gravity Standardisation Net 1971 (IGSN71), Spec. Publ., 4, IAG, Paris,
325  https://apps.dtic.mil/dtic/tr/fulltext/u2/a006203.pdf, 1972.

Moritz, H.: Geodetic reference system 1980, *Bulletin Géodésique*, 54(3), 395–405, 1984.

Moritz, H.: Geodetic Reference System 1980, *Journal of Geodesy*, 74, 128–133, https://doi.org/10.1007/s001900050278, 2000.

Paršeliūnas, E. K. and Petroškevičius, P.: Quality of Lithuanian national gravimetric network, *Harita Dergisi*, 18, 388–392,
330  2007.

Paršeliūnas, E., Obuchovski, R., Birvydienė, R., Petroškevičius, P., Zakarevičius, A., Aksamitauskas, V., and Rybokas, M.: Some issues of the national gravimetric network development in Lithuania, *Journal of Vibroengineering*, 12(4), 683–688, 2010.

Pavlis, N. K., Holmes, S. A., Kenyon, S. C., and Factor, J. K.: The development and evaluation of the Earth Gravitational
335  Model 2008 (EGM2008), *J. Geophys. Res. Solid Earth*, 117, B04406, https://doi.org/10.1029/2011JB008916, 2012.

Petroškevičius, P.: *Influence of gravity field on geodetic measurements (Gravitacijos lauko poveikis geodeziniams matavimams)*, Technika, Vilnius, 290 pp., 2004.

Petroškevičius, P., Paršeliūnas, E. K., Birvydienė, R., Popovas, D., Obuchovski, R., and Papšienė, L.: The quality analysis of the national gravimetric network of Lithuania, *Geodetski vestnik*, 58(4), 746–755, 2014.

340  Reit, B.-G.: On geodetic transformations, *LMV-rapport*, 2010:1, 62 pp., 2009.

Sacher, M., Ihde, J., Liebsch, G., and Mäkinen, J.: EVRF2007 as Realization of the European Vertical Reference System, presented at the Symposium of the IAG Sub-commission for Europe (EUREF), Brussels, 18–21 June, 2008.

Sacher, M. and Liebsch, G.: EVRF2019 as new realization of EVRS, https://evrs.bkg.bund.de/.../EVRF2019_FinalReport.pdf, 2019.

345  Schödlbauer, A.: *Rechenformeln und Rechenbeispiele zur Landesvermessung*, Heft 1, Herbert Wichmann Verlag, Karlsruhe, 145 pp., 1981.

Schödlbauer, A.: *Rechenformeln und Rechenbeispiele zur Landesvermessung*, Heft 2, Herbert Wichmann Verlag, Karlsruhe, 275 pp., 1982.

[Figure]

Šlikas, D., Paršeliūnas, E., Birvydienė, R., and Obuchovski, R. (2025). The countrywide historical gravity dataset of Lithuanian

350    territory [Data set]. Zenodo. https://doi.org/10.5281/zenodo.15090241

Stopar, R.: Map of the Bouguer anomalies, in: *Geological Atlas of Slovenia*, edited by: Novak, M. and Rman, N., Geological

Survey of Slovenia, 20–21, 2016.

The firm „Specgeofizika". Gravimetric map 1:200000, 1965.

Tobler, W. R.: A comparison of spherical and ellipsoidal measures, *The Professional Geographer*, 16(4), 9–12, 1964.

355    Topographic map 1:10 000, www.geoportal.lt, 1958, last access: 14 March 2025.

Torge, W.: *Gravimetry*, Walter de Gruyter, Berlin, New York, 465 pp., 1989.

Wollard, G. P.: New Gravity System – Changes in International Gravity Base Values and Anomaly Values, *Geophysics*, 44(8),

1352–1366, https://doi.org/10.1190/1.1441012, 1979.

Zahorec, P. et al.: The first pan-Alpine surface-gravity database, a modern compilation that crosses frontiers, *Earth Syst. Sci.*

360    *Data*, https://doi.org/10.5194/essd-2020-375, 2020.